# How are families in Germany doing in 2022? Study protocol of a nationally representative, cross-sectional study of parents with children aged 0–3 years

Anna Neumann [1]*, Susanne M. Ulrich[2]*, Frank Sinß[1], Digo Chakraverty[1], Maria Hänelt[1], Ulrike Lux[2], Sarah Ghezih[2], Johanna Löchner[2], Ilona Renner[1]

1 National Centre for Early Prevention, Federal Centre for Health Education, Cologne, Germany,
2 Department of Families and Family Policies, German Youth Institute, National Centre for Early Prevention, Munich, Germany

☯ These authors contributed equally to this work.
* anna.neumann@nzfh.de (AN); ulrich@dji.de (SMU)

## Abstract

### Background

In order for Early Childhood Intervention (ECI) to be effective, data-based information on families' resources, burden and current use of support services for families with young children, as well as on children's health and development is needed. The study *Kinder in Deutschland* [Children in Germany]–*KiD 0–3 2022* aims at providing these data to help us understand families' situation and needs in Germany now, including families' experience of the COVID-19 pandemic.

### Method

The study will recruit up to 300 pediatricians who will invite parents of children aged up to 48 months to participate in the study during a well-child visit. Parents (goal $N = 8,000$) will complete an online-questionnaire with their own web-enabled device. Pediatricians will complete a short questionnaire about each participating family. The questionnaires cover family psychosocial burden and resources, child health and development, use of family support services, as well as the families' experiences of the COVID-19 pandemic. Data will be analyzed to assess patterns of families´ psychosocial burdens and resources, use of support services for families with young children, and children´s health and development. Concordance between parent and pediatrician report will be assessed and comparisons with the predecessor study of 2015 will be drawn.

### Dissemination

Findings will be disseminated through scientific conferences, open access peer-reviewed journals, and dissemination channels of the National Centre for Early Prevention.

**Data Availability Statement:** No datasets were generated or analyzed during the current study.

**Funding:** The study is funded by the Federal Ministry for Family Affairs, Senior Citizens, Women and Youth within the framework of the Federal Foundation for Early Childhood Intervention from the Federal Government's action program "Catching up after Corona for Children and Adolescents". https://www.bmfsfj.de/bmfsfj/aktuelles/alle-meldungen/aktionsprogramm-aufholen-nach-corona-fuer-kinder-und-jugendliche–178422 The funders had and will not have a role in study design, data collection and analysis, decision to publish, or preparation of the manuscript.

**Competing interests:** The authors have declared that no competing interests exist.

## Discussion

The present study will provide parent and pediatrician reports on how families with young children are doing in Germany. These data will be used to inform Germany's early childhood intervention (ECI) program ("Frühe Hilfen") on current needs of families with young children.

## Introduction

Research from disciplines ranging from developmental neuroscience to public health has shown repeatedly that early childhood experiences are of crucial importance for human development [1]. Early adversity is associated with undesirable outcomes related to learning, behavior and physical and mental health and wellbeing, while conversely, positive experiences set children up to become healthy and fulfilled adults [1]. Positive experiences in early childhood crucially include stable and responsive relationships with caregivers, in which children feel safe to explore the world and learn to self-regulate their affect and behavior [2, 3]. Customized early childhood intervention (ECI), targeted at supporting and enhancing caregivers' ability to care for, nurture and teach their child may significantly improve the health and development of children well into adulthood [4]. With its potential to not only increase individual well-being, but to also reduce social and health inequalities [4, 5], ECI is a public health topic of major importance. Germany's ECI program aims at improving psychosocial care for families, promoting equal opportunities for all children to grow up healthy and safe. It offers services to all expectant parents and families with a child aged 0–3 years, with a special focus on families experiencing heightened psychosocial burden [6]. Services embedded in the collaboration between health and child welfare services are free-of-charge and voluntary and include (but are not limited to): pregnancy or child guidance counseling, parenting classes, and home visiting programs by family midwives or nurses [6]. In order for ECI to be effective, we need data-based information on families' resources, burden and current use of support services for families with young children, as well as on children's health and development. The study *Kinder in Deutschland* [Children in Germany]–*KiD 0–3 2022* aims at providing these data to help us understand families' situation and needs in Germany now. Juxtaposition of the 2022 data with data of the study's predecessor study, conducted in 2015, will enable us to show how demographics have changed during the intervening seven years. Two major developments, affecting the entire German population in the years since 2015 cannot be ignored in this regard: the COVID-19 pandemic and the war in Ukraine and their respective repercussions for the German public. Accordingly, parents' experience of the pandemic and worries related to the war will be a special focus of the KiD 0–3 study in 2022.

In the following, the theoretical background of the study is described, including main results of its predecessor study conducted in 2015 (*KiD 0–3 2015*). Details on current developments in Germany as related to the COVID-19 pandemic and the war in Ukraine are given and possible effects on the situation of families with small children are pointed out. After that, the study's aims are specified and details on data collection, data analytic possibilities, and dissemination strategy are provided. We conclude with a discussion of the expected strengths and limitations as well as important implications of the study.

### Child health and development as related to family psychosocial burden and parenting

Human development takes place in multiple layers of ecological systems, ranging from the child's immediate environment to systems affecting the child's life more indirectly, such as the extended families' behavior and well-being, the parents' workplace, community health

services, and finally customs, values and laws of the greater society [7]. In the early years, child health and development largely depend on the families' well-being. Thus, it is important to understand how parents are doing and how their well-being relates to parenting. Parenting in general has been described as "both complex and stressful" [8, p.602], and even more so in psychosocially burdened families. Stress related to parenting may stem from a variety of sources, related to the child (e.g., regulation problems, preterm birth), his/her parent(s), (e.g., mental health difficulties, bonding problems) and the family situation (e.g., single parent, poverty) [7–10]. Children growing up in psychosocially burdened families are more likely to experience mental health and behavioral problems, impaired physical health [11, 12], and often show lower educational attainment compared to their peers growing up in less burdened families [13]. These links are elaborated in greater detail in the family stress model originally proposed to explain the association between economic hardship and child development [14] but since expanded to associations between various stressors, family processes and child development [15]. Based on this expanded model, we assume that family psychosocial burden and resources impact child health and development through parents' mental health and wellbeing, interparental relationship quality and parenting quality.

The present study's predecessor study *KiD 0–3 2015* was a nationally representative German study of N = 8,063 families with children aged 0–3, recruited in pediatric practices [16]. One major finding was that 12.9% of German families with children aged 0–3 years experienced four or more factors related to family psychosocial burden [17], indicating a need for additional support to prevent adverse child health outcomes or maltreatment [18]. Moreover, different groups of burdened families were identified with a latent class analysis, a low burdened group (58,9% of the sample), a highly burdened group (5.2% of the sample) and two groups of families with medium levels of psychosocial burden. While one group was predominantly burdened in the socioeconomic domain e.g., by social welfare receipt or crowded living conditions (18.8%), the other was mainly burdened by parenting stress and familial conflict (17.2%) [19]. In line with international findings, family psychosocial burden was associated with heightened stress experiences of the caregiver [20].

## External factors: The COVID-19 pandemic and the war in Ukraine as influences on the situation of families in

The COVID-19 pandemic and associated contact restrictions, aimed at containing the virus, have been impacting peoples' lives around the globe since spring 2020. In Germany, families had to deal with school and daycare closures, and loss of support with child care [21]. Some parents had to reorganize their professional lives, e.g., by starting to work from home, some were confronted with job loss, job insecurity and associated financial worries and economic hardship [22]. Research from diverse countries has shown an increase in mental health issues in the general public [23], as well as in families with young children [24]. Heightened parenting stress [25, 26], especially in families with younger as compared to older children [27], and increases in youth mental and physical health issues since the pandemic have also been reported, with the most pronounced increases reported for youths living in families with pre-existing vulnerabilities, related to low socioeconomic status, migrant background and restricted housing conditions [28]. Put in general terms, it can be hypothesized, that "Preexisting vulnerabilities within families increase susceptibility to social disruptions and the sequelae of the pandemic, whereas intact or strengthened family well-being will serve to protect children and families from such stressors" [29, p. 632].

Another possible effect of the pandemic might have been the amplification of the *prevention dilemma*: the finding that families with lower levels of psychosocial burden generally use

voluntary preventive services more often than families who would benefit more [30–32], and one of the main challenges for improving psychosocial care for families. During the pandemic, contact opportunities with pediatricians, nursery and daycare staff were greatly reduced for most families. Within Germany's ECI program these professionals have an important role as persons of trust, and they usually inform vulnerable families about the services available and support them in the up-take of suitable services. Besides, due to contact restriction measures, many services could not be offered at all, or at least not as usual during the pandemic [33, 34], which created further barriers in psychosocially burdened families' access to resources and increased social isolation [35]. To allocate resources adequately, it is important to know which families were continuously supported by ECI throughout the pandemic, which services they used and how they evaluate the services used under pandemic circumstances.

At the time of KiD 0–3 2022 data collection (April to November 2022) life had returned to normal in many ways for families in Germany, with strict contact restrictions, including school and daycare closures, no longer in place, and vaccination rates around 75% for the adult population [36]. Nevertheless, high levels of sick leave (due to levels of COVID-19 infections, albeit lower than in 2021, but still widely prevalent especially in the spring and fall of 2022 as well as quarantine regulations for people infected with the virus), continued to impact people's lives. For instance, families with small children might be affected by daycare closures for one to several days due to staff shortage on very short notice.

Just before the start of the field phase, Russia's attack on Ukraine began at the end of February 2022. This had far-reaching consequences for Germany as well: In conjunction with the effects of the Corona pandemic (in particular, the zero-covid policy of the Chinese government), there were supply bottlenecks. A lack of gas supplies from Russia led to fears of an energy crisis. These developments caused a sharp rise in producer prices and thus the highest inflation recorded in Germany since WW II [37, 38]. In addition, there was a profound movement of refugees–Ukrainian women, many with children, sought protection in neighboring countries, including Germany. As of February 2023, over 1 Million war refugees from Ukraine have been registered, almost 70% of them women, and around 357.000 are children under the age of 18 [39], with additional pressure on the housing market as one consequence. Moreover, there was a subjective and objective threat situation for Germany (threat of nuclear strikes by the Russian leadership, fear of a major European or world war). Therefore, it was decided at rather short notice to assess the influence of this crisis on the study population.

## Research questions and objectives

Taken together, drawing on the bio-ecological systems model of human development and the family stress model, on the *KiD 0–3 2015* study, as well as on recent empirical findings on family well-being during and since the pandemic, *KiD 0–3 2022* aims to establish a reliable knowledge base on families' psychosocial burdens, service utilization, and children's health outcomes in Germany in 2022. Specifically, we aim to answer the following overarching research questions:

**RQ1**: What is the proportion of psychosocially burdened families, what are the most common burdens and resources and how are they distributed in families from different socioeconomic backgrounds? How do these numbers compare to numbers from 2015?

**RQ2**: How do parents and pediatricians evaluate children's well-being and development and how do family psychosocial burden and wellbeing relate to child well-being and development?

**RQ3**: Which experiences (both positive and negative) of the pandemic do families report the most and what are parents' and pediatricians' views on the impact of the pandemic on child development?

**RQ4**: Which family support services are known and used by families? Do families with different levels of psychosocial burden differ in the services they know and use? Do parents rate the services they did use as helpful?

A great asset and stand-alone feature of the *KiD 0–3* study is that data is collected during well-child visits to the family's resident pediatrician. Nearly all parents (99%) in Germany make use of these visits in the first two years of their child's life, and a still very high percentage of 93% does so with their three-year-old child [40]. This enables us to collect data of families from diverse socioeconomic backgrounds [41]. Accordingly, the knowledge gained in the *KiD 0–3 2022* study will be of great use to further develop early childhood interventions tailored to families' actual needs.

## Materials and methods

### Design

The *KiD 0–3 2022* study aims to include about 8,000 families with children aged up to 48 months who visit their resident pediatrician for a regular well-child visit (U3-U7a screening). Resident pediatricians will be recruited as *study centers*. In short, in their pediatricians' practice, parents will be informed about the study by practice staff and/or their pediatrician. Parents who agree to participate will complete a written informed consent form, before they receive an invitation with a QR-Code and an identification number (ID) to fill out an online questionnaire on psychosocial burden, resources and service use. After the well-child examination, the attending pediatrician completes a short questionnaire on the family by using the same ID, allowing linkage of family and pediatric data. The field phase, which will be conducted by the research institute House of Research (HoR) (https://www.house-of-research.de/) under supervision by the authors, is planned to run from April to December 2022.

### Sample size

Based on experiences from the *KiD 0–3 2015* study, we estimate to achieve a target size of up to 300 pediatricians by contacting 1,500 pediatricians with an estimated response rate of 15%. This estimate is based on the KiD 0–3 2015 study, for which non-responder analyses showed a satisfactory distribution of key criteria, including region and type of practice (single or group practice). In KiD 0–3 2022, we will also conduct in-depth analyses on the distribution and, if necessary, include these in the weighting strategy. The goal is for each pediatrician to recruit 25 families on average to participate in the study. With a sample size of 8,000 families, we will be able to draw reliable statements about the population starting at a very low prevalence of 0.5% in the sample. This will allow us to capture even rare psychosocial stress factors (e.g. child disability) and to analyze special subgroups.

### Survey sample

Resident pediatric practices (both individual and joint) will be eligible to participate if they conduct well-child visits.

Eligibility criteria for parents are: 1) the parent(s) visit(s) their child's pediatrician (who participates in the study) for a well-child examination (U3-U7a Screening), 2) the parent is either a biological or adoptive parent of the child, 3) the child's age is < = 48.0 months, 4) the child

resides, at least temporarily, with the accompanying parent, 5) the parent is able to understand and read one of the five languages, in which the documents and parent questionnaire are provided (German, English, Turkish, Arabic, or Russian), and 6) the parent is able to use his/her smartphone or another internet-eligible device to complete the online-questionnaire (current studies show that 98% of the expected age groups have an internet-eligible smartphone [42]).

## Measures/survey development

**Parent questionnaire.** The parent questionnaire comprises information on family psychosocial stress and resources in different domains (child, parent, family and household situation), questions on the personal experiences of the COVID-19-pandemic and on the knowledge, use and acceptance of prevention and support services for families with young children. It is based on its 2015 predecessor study [19]. The selection of psychosocial stress characteristics was strongly guided by the results of previously published meta-analyses [43, 44] and valid instruments for child maltreatment, parenting problems and child development difficulties [45]. Items from the questionnaire are displayed in Table 1.

**Pediatrician questionnaire (family-specific).** The pediatrician questionnaire comprises a general assessment of child health, age-appropriate development and a general assessment of family psychosocial stress, as well as an assessment of the possible impact of the COVID-19 pandemic on the physical, social and affective development of the child. Items and their wording are (in part) based on the booklet used by pediatricians for the documentation of the well-child visits and allow the juxtaposition with parent report (see Table 2). The pediatrician questionnaire was developed in close cooperation with the professional association of pediatricians and pretested with two resident pediatricians. Pediatricians may choose to complete the questionnaires online or on paper.

**General practice documentation.** At the end of the field phase, participating pediatricians will be asked to complete an additional short questionnaire, with questions regarding characteristics of the pediatric practice, cooperation with local early childhood intervention networks, estimated proportions of families visiting the practice with psychosocial burden and further need for support. Other questions regard a general evaluation of the impact of the COVID-19-pandemic on families and child development. Finally, items on possible effects of the Ukraine war on families' psychosocial burden and on work in the pediatric practices, are included.

## Recruitment of resident pediatricians as study centers

Resident pediatricians are chosen by a proportionally stratified random sample of 1,500 pediatricians drawn from the address database *direkt+online* (https://www.direktundonline.de/), which consists of publicly available data and includes almost all of German resident pediatric practices (round about 5,700 addresses). Stratification criteria for the pediatric practice sample were the 16 German federal states, type of pediatric practice (individual or joint) and community size (more or less than 100,000 inhabitants). All 1,500 practices drawn from *direkt+online* received invitations and an expression of interest by mail. Non-responding practices are contacted by the field institute by phone. Due to deviations from the quota plan, additional pediatric practices from an additional proportionally stratified random sample of 2,500 pediatricians, drawn from the *direkt+online* database, in underrepresented areas are being contacted.

Pediatricians willing to participate in the study receive a package containing all documents needed for implementing the study in their practice. These packages include documents to inform parents about the study and support the recruitment of parents as well as the paper-

**Table 1. Overview of constructs and related items of the parent questionnaire.**

| Construct | Items | Nr. of Items | Scale | Reference/Source |
|---|---|---|---|---|
| **Child Characteristics** | | | | |
| Eligibility Criteria | Relationship to the child | 1 | Mother/Father | KiD 0–3 2015 |
| | Well-child Screening | 1 | U3-U7a | self-constructed |
| | Child living in the household | 1 | Yes/No/Partly | KiD 0–3 2015 |
| Sociodemographic | Child age | 1 | 0–48 months | KiD 0–3 2015 |
| | Child sex | 1 | Male/Female/Diverse | self-constructed |
| Pregnancy | Planned pregnancy | 4 | Yes/No | KiD 0–3 2015 |
| | Abortion/adoption thoughts | | | KiD 0–3 2015 |
| | Smoking during pregnancy | | | KiD 0–3 2015 |
| | Regular check-ups during pregnancy | | | KiD 0–3 2015 |
| Perinatal adversities | Premature birth before 37. week | 1 | Yes/No | KiD 0–3 2015 |
| | Low birth weight <2,500 g | 1 | Yes/No | KiD 0–3 2015 |
| | Breastfeeding | 1 | Yes/No | KiD 0–3 2015 |
| | Duration of breastfeeding | 1 | 0–48 months | KiD 0–3 2015 |
| Child regulatory problems | Screaming and crying behavior | 3 | Yes/No | Wessels Index [46], KiD 0–3 2015 |
| | Experienced burden due to child's crying | 1 | 1–4 | KiD 0–3 2015 |
| | Experienced burden due to child's sleeping behavior | 1 | 1–4 | KiD 0–3 2015 |
| | Experienced burden due to child's eating behavior | 1 | 1–4 | KiD 0–3 2015 |
| Child temperament | Negative emotionality | 5 | 1–4 | SGKS [47] |
| **Parent Characteristics** | | | | |
| Parenting behavior | Parental responsiveness | 5 | 1–6 | CECPAQ [48] |
| Parenting stress | Doubts in parenting competence and low parental sensitivity | each 2 | 1–5 | German PSI [49] |
| Attribution of the child | Negative attribution/tendency to harsh parenting | 2 | 1–4 | EMKK [50] |
| Impulsivity | Feelings of inner anger | 1 | 1–4 | CAPI [51] |
| Parental burnout | Feelings of being very exhausted | 1 | daily, 1–2 p.w., seldom/never | BPBS [52] |
| Depression/Anxiety | Sings of depression or anxiety symptoms | 4 | 1–4 | PHQ-4 [53] |
| Postnatal depression | Symptoms of depression/anxiety after birth | 1 | no, at least for 2 weeks, most time | self-constructed |
| Adverse childhood experiences | Much love or harsh punishment in parent's childhood | 3 | 1–4 | EMKK [50] |
| **Social support/ parental partnership** | | | | |
| Social support | Persons available for child care or to give advice on parenting problems | 2 | 1–4 | p29m [54] |
| Child care | Other parent, daycare center, daycare mother, grand-parents or other | 5 | Yes/No | self-constructed |
| Partnership satisfaction | How happy in general with partnership | 1 | 1–10 | pairfam [55] |
| Loud and frequent quarrels | Often severe argument with partner | 1 | 1–6 | KINDEX [56] |
| Co-parenting | Seeking solutions together, often differences in opinion | 2 | 1–5 | Co-parenting Scale [57] |
| **Family and household situation** | | | | |
| Household composition | Nuclear family or single parent or new parent | 3 | Yes/No | KiD 0–3 2015 |
| Young mother (<21 years) | Age at birth of the child | 1 | In Years | KiD 0–3 2015 |
| Number of children in household | Direct questioning on number and age of household members | 2 | | KiD 0–3 2015 |
| Migration background | Birthplace of at least one parent or child is not Germany | 3 | Yes/No | Migration regulation act §6, KiD 0–3 2015 |

*(Continued)*

**Table 1.** (Continued)

| Construct | Items | Nr. of Items | Scale | Reference/Source |
|---|---|---|---|---|
| **Child Characteristics** | | | | |
| Social welfare receipt | Financial support for at least one family member in the last 12 months | 1 | Yes/No | KiD 0–3 2015 |
| Level of education | Highest professional degree and highest level of school qualification | 2 | None/completed apprenticeship/professional degree/master/university degree None/lower secondary/university entrance/high school diploma | ISCED [58] |
| **COVID-19 pandemic** | | | | |
| Pregnancy during corona | Negative or positive feelings | 1 | 1–4 | self-constructed |
| Worries and fears | During the time of pandemic and today | each 10 | 1–5 | self-constructed |
| Positive aspects | During the time of pandemic and today | 8 | 1–5 | self-constructed |
| Influence on child development | Physical, socio-emotional, mood | 3 | No, positive, negative | self-constructed |
| Overall evaluation of the pandemic | Personal, family, positive aspects | 3 | 1–5 | pairfam [55] |
| **Service uptake** | | | | |
| Knowledge and use of universal services | Medical services e.g. Prenatal classes, medical services after childbirth, midwife assisting for first 8 weeks (covered by statutory health insurance) and social services e.g. parenting courses, parent-child-group | 18 | Yes/No | KiD 0–3 2015 |
| Knowledge and use of (rather) selective consultation services | Child guidance center, pregnancy counseling, specialized counseling (e.g. crying problems), early intervention | | | |
| Knowledge and use of early childhood interventions | Home-visiting programs, welcome visits, volunteer visits | | | |
| Contact to child welfare services | Family preservation programs offered by child protection/child welfare service | 1 | Yes/No | KiD 0–3 2015 |
| Evaluation of the services | Question on how the services used are evaluated | max. 18 | 1–5 | self-constructed |
| Personal barriers of service uptake | Questions on personal attitude | 3 | 1–4 | NZFH-Accessibility study [59] |
| **Ukraine crisis** | Worries and fears e.g. financial worries and experienced burden | 6 | 1–4 | self-constructed |

pencil pediatrician questionnaires (if the pediatrician chose not to complete the questionnaires online), and information about how to implement the study in the pediatric offices. Pediatricians and their staff are urged to study the *how to*-material provided before starting to approach parents to participate. These materials include a 17-minute training video and a detailed manual, to be found on a password-protected study website, as well as video-conference trainings, provided by members of the study team.

## Recruitment of parents and study procedure

Eligible parents will be invited to participate in the study by the practice staff or by the pediatrician him- or herself. They will receive a letter of invitation and a description of the study including information on data protection and privacy. Parents willing to participate complete an "informed consent" form and completed the questionnaire voluntarily and anonymously according to the Declaration of Helsinki [60]. The signed version of the form will be kept in a secure place in the pediatric practice and later mailed to the research institute HoR, while

**Table 2. Overview of the pediatrician questionnaire.**

| Construct | Items | Nr. of Items | Scale |
|---|---|---|---|
| Child health | Overall assessment of child's health | 1 | 1–5 |
| | Chronic disease or disability | 1 | Yes/No |
| Child development | physical, social and affective development | 3 | 1–3 |
| Child regulatory problems | dichotomous<br>regulatory problems related to crying, sleeping, feeding/eating, playing<br>Need for support related to Regulatory problems | 1<br>4<br>1 | Yes/No<br>1–5<br>1–5 |
| Parent-child interaction | Overall assessment<br>Need for support | 1<br>1 | 1–5<br>1–5 |
| Family psychosocial burden | Overall assessment<br>Need for support | 1<br>1 | 1–3<br>1–3 |
| Recommendation / referral to support services | Medical support<br>Psychosocial support | 2 | Yes/No |
| Possible consequences of the COVIC.19 pandemic on child development | Physical development<br>Social development<br>Affective development | 3 | Yes/No, positive/yes/negative /don't know |

*Note.* Items are self-constructed, based on the well-child screening booklets and pre-tested with pediatricians.

parents may keep a copy of the form. Parents start the online questionnaire via a QR-code on the informed consent form on their own smartphone or other device using a unique ID provided by the practice staff. Ideally, parents will complete the questionnaire (approx. 25 minutes) in the pediatric practice. If necessary, parents can pause the questionnaire and complete it later by reentering their password. Participating families will receive a voucher worth € 20, which can be redeemed in a variety of online stores. The recruitment and study procedure are described in Fig 1.

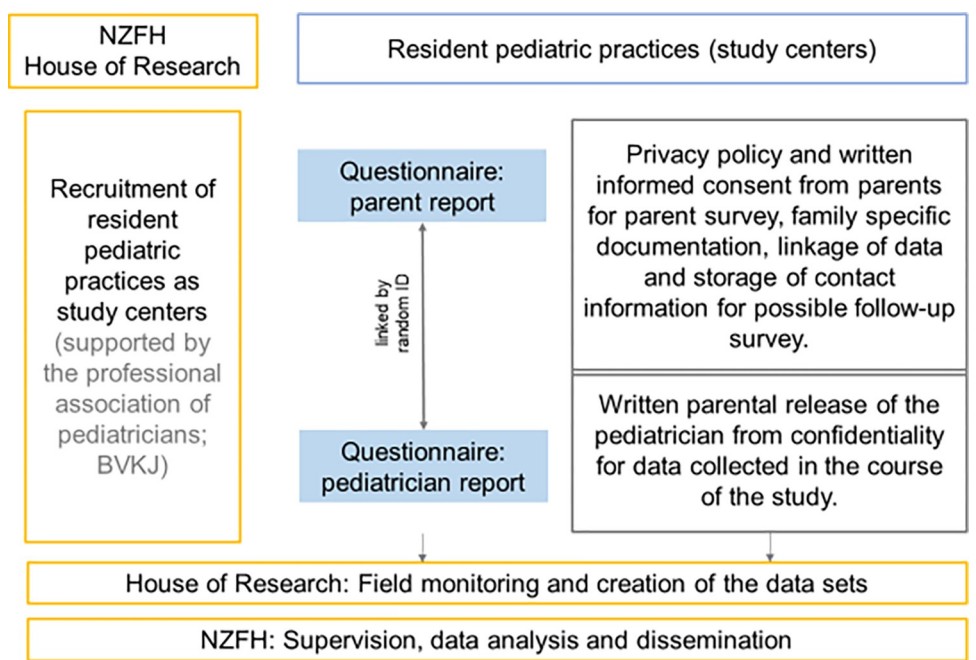

**Fig 1. Recruitment and study procedure.** NZFH = National Centre for Early Prevention.

Pediatricians complete the pediatric questionnaire for each participating family after the well-child screening. By entering the family's individual ID, family and physician data can be linked. Physicians have no insight into the answers provided by the parents on the parent questionnaire.

Study and response monitoring are conducted by the field institute which also maintains a telephone hotline for the pediatricians. Once pediatricians will have gathered complete data on 30–35 families (or earlier, if they choose to do so) and the research institute HoR has received all family informed consent forms (and pen-and-paper pediatrician questionnaires, if applicable), pediatricians will be asked to complete and return the final pediatrician questionnaire (pen-and-paper). Pediatricians will receive a compensation of 60 € for each successfully recruited family (complete parent and pediatrician questionnaire, signed parental informed consent form), an amount which about doubles the pay pediatricians receive for a standard well-child screening.

## Data protection and ethical approval

Participants provide written informed consent. The consent form for the families includes information on data protection and security. It also includes and information on participants' rights regarding their data and on how to execute these rights (e.g., contact information in case the consent to participate wants to be withdrawn), a confirmation of the release from medical confidentiality and a query in interest for a follow-up study. The concept and all documents were approved by the National Center for Health Education's data protection officer and the ethical committee of the German Youth Institute (Ethikkommission des Deutschen Jugendinstituts e.V., Ethics Vote KiD 0–3 2022, request number 2021/005; approval granted on 27 January 2022)

## Patient and public involvement

Regarding *KiD 0–3 2022*, study team are in close contact with the Professional Association of Pediatricians (Berufsverband der Kinder- und Jugendärzte, BVKJ). The BVKJ plays an important role in publicizing information about, and rallying for support for the study among its members. The study procedure was pretested in three pediatric practices. In addition, psychosocially burdened mothers pretested the parent questionnaire and provided feedback on its comprehensibility and content relevance.

## Data analysis plan

The dataset will include linked parental self-report and pediatrician-report data on each family. A combination of design and poststratification weights will be used to adjust for the German federal states and for several relevant sociodemographic characteristics (age, citizenship, education and vocational training of the parents as well as the household constellation) drawn from the latest version of the German Microcensus. The weighted and plausibility checked data will then be used to answer our research questions. For the analysis, we plan to use only complete cases for which were the data set from the families and the pediatrician is completed. We also plan a non-reponder analysis for families for whom only data from the pediatricians (and the families informed consent) is available, but who have not completed the parent questionnaire. In particular, the following analyses are planned:

**RQ1: Analysis of the distribution of familial psychosocial burden and resources.** To examine the distribution of familial psychosocial burden and resources, prevalence rates of individual stressors and resources (e.g., risk for poverty, social support) as well as the proportion of families with a cumulation of psychosocial stressors will be determined. Furthermore,

expanding the latent class model for screening and identifying psychosocial burdened families described by Lorenz et al. [19], we will include familial resources along with the familial stressors. The results will be juxtaposed with results of the *KiD 0–3 2015* study e.g., by comparing prevalence rates and proportions of psychosocially burdened families.

**RQ2: Analysis of child health and development.** We will analyze similarities and differences between pediatrician and parent assessment of child health and development. We plan to conduct subgroup analyses to examine if difficult circumstances (e.g. high levels of burden or certain child, parent or family related stressors in particular) are particularly associated with child health problems or not age-appropriate child development. Multivariate analyses are planned to identify determinants of health and delayed development.

**RQ3: Analysis of the experience of the COVID-19 pandemic and the war in Ukraine.** We will analyze which experiences during the pandemic are reported most often by parents, and how these experiences relate to socioeconomic characteristics such as poverty and low levels of formal education. Parents' and pediatricians' impressions of the impact of the pandemic on child health and development will be compared. Additionally, we will present descriptive data on respondents' worries and concerns related to the war in Ukraine for the entire sample and for subgroups such as socioeconomically burdened families.

**RQ4: Analysis of the use of prevention and support services.** We will examine the service uptake of our sample and compare the knowledge and use of the support services between different target groups (e.g. families in poverty, mentally burdened or single parents) to understand which families are reached by support services and how families evaluate these services. Multivariate analyses are planned in order to identify determinants of service uptake. Data on the use of support services will be compared to respective data from the *KiD 0–3 2015* study.

## Dissemination of findings

Results of the study will be made available to practitioners, policy makers and the broader public by the NZFH's in-house publishing (German booklets, newsletters, homepage, press releases etc.; see *www.fruehehilfen.de*). Additionally, findings will be reported in international scientific journals and at scientific conferences in the areas of public health, child development, pediatrics, family and youth services and related areas.

## Discussion

The *KiD 0–3 2022* study will provide nationally representative data on psychosocial burden and resources, as well as on the uptake of support services designed for families around the birth of a child and up to age three in Germany. Current date data on families with young children regarding health and development are sorely needed for the advance of Germany's early childhood intervention program. This is especially important due to the challenges many families are facing since the pandemic, and additional worries families in Germany might have in relation to the war in Ukraine.

Data collection will take place during well-child visits in resident pediatric practices, which is a particular strength of the planned cross-sectional study. Since nearly all families (about 99%) in Germany make use of these visits [40], and in most cases, parents bestow great trust upon their pediatrician, this study design has the power to bypass the socioeconomic bias inherent in studies with similar populations, as shown by the *KiD 0–3 2022* predecessor study in 2015 [41]. Since the professional association of pediatricians supports the *KiD 0–3 2022* study, it is likely, that a representative sample of resident pediatricians can be drawn. Representativeness will be determined by comparison with the latest microcensus data. By the collection of pediatrician reports on each family willing to participate, self- and other report will

be available, reducing effects of reporter bias. Another strength of the planned study is its expected sample size of 8.000 families, which will allow us to capture prevalence rates of factors that are relatively rare but associated with high levels of burden. Finally, juxtaposition of the data with those from the predecessor study (with a representative sample of 8.063 families) from 2015, will allow us to research possible changes in demographics over the years.

Results of the study are highly relevant to inform the further development of Germany's early childhood intervention program. We need to understand which families are reached by which programs, and if and how the COVID-19 pandemic had an impact on the uptake of (digital) support measures. The results will help to improve current prevention/intervention programs, make potential adjustments to their accessibility and develop further programs, tailored to families' needs. Additionally, it will be important to explore whether the expansion of Germany's Early Childhood Intervention program between 2015 and 2022 led to changes in how many and which families know about and make use of the associated support measures.

Beyond practical implications for early childhood intervention, we expect results of the study to contribute to a better understanding of the complex interplay between family psychosocial stressors and resources in this specific challenging period of contact restrictions and global threat and how they relate to child health and development in the important and formative early years. Also, to our knowledge, our study will be the first to provide data on the well-being of young children and their families in Germany since the COVID-19 pandemic in a nationally representative sample. The same is true for information on parental worries related to the Ukraine crises. Also, pediatrician-reported data on potentially increased workload in resident pediatrician offices related to the pandemic and to the need to serve refugees from the Ukraine, among which there are many mothers with children, will also be provided. We expect all of these data to be of great relevance to policy makers in order to better prepare for future crises, putting families and children first.

A major strength of the KiD 0–3 study, is that data from 2022 can be compared to data from 2015. However, regarding questions on the experience of the COVID-19 pandemic by families with young children, changes in prevalence rates between 2015 and 2022 cannot be attributable to the sole impact of the COVID-19 pandemic. It is important to note, that the KiD 0–3 2022 study will not provide data on the "live" experience of the COVID-19 pandemic during times of the strictest contact restrictions. However, what will be assessed is families experience of the months and years since the start of the pandemic, most of it in retrospect. Since COVID-19 pandemic related stress can be expected to build up over time, we consider this perspective highly relevant to our research question of "how families in Germany are doing *now*". Data cannot only be compared across time, but also across reports (parents and pediatricians). Due to time restrictions, the pediatrician questionnaire had to be kept as short as possible, and consequently pediatric statements on child health and development and on family psychosocial burden are rather general in nature.

## Conclusion

Taken together, the *KiD 0–3 2022* will provide data of great relevance to the fields of family research, public health, child development and pediatric practice. In addition to further our knowledge on the pattern of family psychosocial burden and resources and their association with child health and development in the critical early years of childhood, we expect to learn a lot about the experience of families with young children of the COVID-19 pandemic. And last but not least, the data will be of great use for the further development of early childhood intervention, thereby ultimately supporting parents in enabling their child to have the best possible start in life.

## Acknowledgments

We wish to thank the families who took part in the studies and the pediatrician practices who served as study centers. We also want to thank the Professional Association of Pediatricians (BVKJ e. V.) who supports this study with commitment.

## Author Contributions

**Conceptualization:** Anna Neumann, Susanne M. Ulrich, Ilona Renner.

**Investigation:** Anna Neumann, Susanne M. Ulrich, Frank Sinß, Digo Chakraverty, Maria Hänelt, Ulrike Lux, Sarah Ghezih, Johanna Löchner, Ilona Renner.

**Methodology:** Anna Neumann, Susanne M. Ulrich, Frank Sinß, Digo Chakraverty, Maria Hänelt, Ulrike Lux, Sarah Ghezih, Johanna Löchner, Ilona Renner.

**Project administration:** Anna Neumann, Susanne M. Ulrich, Ulrike Lux, Johanna Löchner, Ilona Renner.

**Resources:** Anna Neumann, Susanne M. Ulrich, Frank Sinß, Digo Chakraverty, Maria Hänelt, Ulrike Lux, Sarah Ghezih, Johanna Löchner.

**Supervision:** Ilona Renner.

**Writing – original draft:** Anna Neumann, Susanne M. Ulrich.

**Writing – review & editing:** Frank Sinß, Digo Chakraverty, Maria Hänelt, Ulrike Lux, Sarah Ghezih, Johanna Löchner, Ilona Renner.

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
