## [Decision Letter · Decision Letter 0]

25 Jan 2023

PONE-D-22-25670How are families in Germany doing since the COVID-19 pandemic? Study Protocol of a nationally representative, cross-sectional study of parents with children aged 0-3 yearsPLOS ONE

Dear Dr. Anna,

Thank you for submitting your manuscript to PLOS ONE. After careful consideration, we feel that it has merit but does not fully meet PLOS ONE’s publication criteria as it currently stands. Therefore, we invite you to submit a revised version of the manuscript that addresses the points raised during the review process.

We look forward to receiving your revised manuscript.

Kind regards,

Humayun Kabir, MSc in Epidemiology

Academic Editor

PLOS ONE

Journal Requirements:

2.Please include your ethics statement in the Methods section of your manuscript. In the Methods section of your revised manuscript, please include the full name of the institutional review board or ethics committee that approved the protocol, the approval or permit number that was issued, and the date that approval was granted.

"The KiD 0-3 2022 study is conducted by the National Center for Early Prevention (NZFH) and funded by the Federal Ministry for Family Affairs, Senior Citizens, Women and Youth (BMFSFJ) within the framework of the Federal Foundation for Early Childhood Intervention from the Federal Government's action program "Catching up after Corona for Children and Adolescents".  The NZFH was established in 2007 to steer and provide monitoring and support for the ECI program, and contributes to health promotion and preventive child protection by supporting families. It is hosted by the Federal Centre for Health Education in collaboration with the German Youth Institute. "

"The study is funded by the Federal Ministry for Family Affairs, Senior Citizens, Women and Youth within the framework of the Federal Foundation for Early Childhood Intervention from the Federal Government's action program "Catching up after Corona for Children and Adolescents”. 

https://www.bmfsfj.de/bmfsfj/themen/corona-pandemie/aufholen-nach-corona

The funders had and will not have a role in study design, data collection and analysis, decision to publish, or preparation of the manuscript."

Reviewers' comments:

Reviewer's Responses to Questions

**Comments to the Author**

1. Does the manuscript provide a valid rationale for the proposed study, with clearly identified and justified research questions?

Reviewer #1: Partly

2. Is the protocol technically sound and planned in a manner that will lead to a meaningful outcome and allow testing the stated hypotheses?

Reviewer #1: Partly

3. Is the methodology feasible and described in sufficient detail to allow the work to be replicable?

Reviewer #1: Yes

4. Have the authors described where all data underlying the findings will be made available when the study is complete?

Reviewer #1: Yes

5. Is the manuscript presented in an intelligible fashion and written in standard English?

Reviewer #1: Yes

6. Review Comments to the Author

You may also provide optional suggestions and comments to authors that they might find helpful in planning their study.

Reviewer #1: Thank you for the opportunity to review this paper, which describes a Protocol for a nationally representative, cross-sectional study of parents with children aged 0-3 years in Germany. My questions mostly relate to the framing of the study including the aims, and how these will be investigated.

1. The primary rationale for this Protocol is understanding the experience of the COVID-19 pandemic. The secondary rationale is a more general investigation of the health and social circumstances of families in Germany now, and as relates to the previous survey (KiD 0-3 2015). As the authors note in the Discussion, it is not possible to attribute changes in this time (2015-22) to the pandemic alone. Because of the challenges inherent in attributing causality to repeated cross-sectional data, especially one over a long period of time, and with an intervening crisis, I query whether the protocol should be framed with regards to the experience of COVID-19 at all. Similarly, how will it be possible to understand the potential gains or loss of the early intervention efforts (ECI) over this time? In thinking about why the authors have chosen his framing, I wonder if the resourcing for the study was acquired for the purpose of investigating the experience of the pandemic. I recommend reorganizing the paper, so it aims to:

o Understand the circumstances of families with young children now, enabling comparisons with the 2015 data to understand (a) who needs what and (b) whether the demographics/experiences have changed in the intervening 7 years.

o Explore families’ experiences of the (a) pandemic and (b) service use over the intervening (number-please specify) years, as a way of speculating what did and didn’t make a difference.

2. If the authors retain the current framing whereby the study is intended to measure the impact of the COVID-19 pandemic, the following need addressing:

o Introduction: providing information about Germany’s experience of the pandemic and public health restrictions over the 3 years since it began.

o Throughout: there is a high risk of recall bias in the design of the study, both as relates to the pandemic and ECI. Please explain how this will be mitigated. Please also specify the periods of time each measure relates to.

3. In the analysis section, please explain which analytic methods will be used to test the aims/make comparisons. Please also explain how missing data will be addressed for the different sets of analysis. I note that, if the study is reframed as in (1) above, descriptive analyses would be sufficient to address the aims.

Other comments:

4. Line 104: I don’t understand this paragraph: different types of burden and stress are used multiple times and I can’t tell how they are different between the groups.

5. Line 123: “research has demonstrated changes” – this reads as though they are sustained changes, but this concept goes beyond the scope of this paper and the study cited was from the early months of the pandemic.

6. Line 156: the phrasing “in the face of the COVID-19 pandemic” – it is not possible to judge whether this is current for German families or not, without more information in the Introduction. I have assumed like many countries, we’re in a ‘post-COVID-19’ world, in the sense that lockdowns have finished, people are vaccinated, and there’s a lot of COVID-19 circulating.

7. Line 193: when is consent invited?

8. Line 202: The 15% uptake through pediatricians is concerning, though understandable given the high research requirements in the face of overburdened health systems. I see that weighting will be used to approximate the German population; are there any risks to consider here around the representativeness of those attending pediatric offices and will this design need to consider those who don’t attend?

9. Line 199: Has recruitment already finished (November 2022)?

10. Table 1: why did the authors design measures for the pandemic experience instead of using questions from other well-known studies, like the UK Young Minds Matters or Australia’s National Child Health Poll or CRISIS questionnaire?

11. How feasible is the 25-minute questionnaire in clinic, plus the administrative parts, e.g. are clients waiting that long and if they are, will embedding this research increase these times (i.e. if the pediatrician must also enter data)?

12. How much would a pediatrician be paid for a standard consult, and how does the 60 euro compare? This also helps understand feasibility.

13. Line 376: first data – please add ‘In Germany’.

14. Line 377: The aspects on the experience of the Ukraine war need more justification if you want to include them as it was a surprise to see them here.

15. The PLOS ONE data sharing for Protocols asks: “For protocols without pilot or preliminary data, authors are strongly encouraged to state how they plan to share research data from their study when it is completed or published.” Can you please add something to this?

7. PLOS authors have the option to publish the peer review history of their article (what does this mean?). If published, this will include your full peer review and any attached files.

Reviewer #1: **Yes: **Anna Price

<quillbot-extension-portal></quillbot-extension-portal>

---

## [Author Response · Author response to Decision Letter 0]

21 Mar 2023

Dear Editor, dear Reviewer,

Thank you for your thorough reading of our manuscript and your very helpful comments. We are very grateful for the opportunity to resubmit our manuscript. As detailed below, we made several changes to the manuscript, following your suggestions and comments. We repeat each issue raised and state how we dealt with it. We hope, that you agree with us, that these changes added to the quality of our manuscript and we look forward to hearing from you. 

On behalf of all authors, 

Anna Neumann

 We double-checked the manuscript for style requirements. 

2.Please include your ethics statement in the Methods section of your manuscript. In the Methods section of your revised manuscript, please include the full name of the institutional review board or ethics committee that approved the protocol, the approval or permit number that was issued, and the date that approval was granted.

We included the information requested in the section “Data protection and ethical approval”. 

"The KiD 0-3 2022 study is conducted by the National Center for Early Prevention (NZFH) and funded by the Federal Ministry for Family Affairs, Senior Citizens, Women and Youth (BMFSFJ) within the framework of the Federal Foundation for Early Childhood Intervention from the Federal Government's action program "Catching up after Corona for Children and Adolescents". The NZFH was established in 2007 to steer and provide monitoring and support for the ECI program, and contributes to health promotion and preventive child protection by supporting families. It is hosted by the Federal Centre for Health Education in collaboration with the German Youth Institute. "

"The study is funded by the Federal Ministry for Family Affairs, Senior Citizens, Women and Youth within the framework of the Federal Foundation for Early Childhood Intervention from the Federal Government's action program "Catching up after Corona for Children and Adolescents”. 

https://www.bmfsfj.de/bmfsfj/themen/corona-pandemie/aufholen-nach-corona

The funders had and will not have a role in study design, data collection and analysis, decision to publish, or preparation of the manuscript."

We deleted the section “Funding” from the manuscript. An amended statement is included in the cover letter. It reads as the Funding Statement above. 

 Thank you for bringing this important topic up again. We would like to include the following statement related to data availability: 

“The data in this study is collected from patients and their physicians. The patients provide informed consent to release their medical information for the purpose of our analysis. However, to protect the privacy and confidentiality of the patients, the data is not publicly available. The authors confirm that all data necessary to support the findings of this study are available upon request to the corresponding author. Requests for access to the data should be made to the corresponding author and will be subject to ethical review and approval.”

We deleted information regarding the ethics statement from the abstract. 

Reviewers' comments:

Reviewer's Responses to Questions

Comments to the Author

1. Does the manuscript provide a valid rationale for the proposed study, with clearly identified and justified research questions?

Reviewer #1: Partly

2. Is the protocol technically sound and planned in a manner that will lead to a meaningful outcome and allow testing the stated hypotheses?

Reviewer #1: Partly

3. Is the methodology feasible and described in sufficient detail to allow the work to be replicable?

Reviewer #1: Yes

4. Have the authors described where all data underlying the findings will be made available when the study is complete?

Reviewer #1: Yes

5. Is the manuscript presented in an intelligible fashion and written in standard English?

Reviewer #1: Yes

6. Review Comments to the Author

You may also provide optional suggestions and comments to authors that they might find helpful in planning their study.

Reviewer #1: Thank you for the opportunity to review this paper, which describes a Protocol for a nationally representative, cross-sectional study of parents with children aged 0-3 years in Germany. My questions mostly relate to the framing of the study including the aims, and how these will be investigated.

1. The primary rationale for this Protocol is understanding the experience of the COVID-19 pandemic. The secondary rationale is a more general investigation of the health and social circumstances of families in Germany now, and as relates to the previous survey (KiD 0-3 2015). As the authors note in the Discussion, it is not possible to attribute changes in this time (2015-22) to the pandemic alone. Because of the challenges inherent in attributing causality to repeated cross-sectional data, especially one over a long period of time, and with an intervening crisis, I query whether the protocol should be framed with regards to the experience of COVID-19 at all. Similarly, how will it be possible to understand the potential gains or loss of the early intervention efforts (ECI) over this time? In thinking about why the authors have chosen his framing, I wonder if the resourcing for the study was acquired for the purpose of investigating the experience of the pandemic. I recommend reorganizing the paper, so it aims to:

o Understand the circumstances of families with young children now, enabling comparisons with the 2015 data to understand (a) who needs what and (b) whether the demographics/experiences have changed in the intervening 7 years.

o Explore families’ experiences of the (a) pandemic and (b) service use over the intervening (number-please specify) years, as a way of speculating what did and didn’t make a difference.

2. If the authors retain the current framing whereby the study is intended to measure the impact of the COVID-19 pandemic, the following need addressing:

o Introduction: providing information about Germany’s experience of the pandemic and public health restrictions over the 3 years since it began.

o Throughout: there is a high risk of recall bias in the design of the study, both as relates to the pandemic and ECI. Please explain how this will be mitigated. Please also specify the periods of time each measure relates to.

We greatly appreciate the reviewer’s thorough reading of our manuscript and would like to thank her for her suggestions to reframe the rationale for the study. We agree with the reviewer, that the best way to frame the study, allowing for the most clarity and straightforwardness in leading up to the research questions as well as in data analyses and interpretation, is to focus on understanding the circumstances of families with young children in Germany now in order to understand who needs what and to see how demographics changed in comparison to data from the 2015 study. As suggested, we will then treat questions related to the experience of the pandemic and the service use over the intervening seven years as exploratory. Accordingly, we reorganized and rewrote the introduction: 

- It now starts with a paragraph on the importance of the early years for human development, on how early childhood intervention (ECI) may help in creating more equal opportunities for all children and a description of Germany’s ECI program. The paragraph concludes with the introduction of the KiD 0-3 study and its aim and gives the reader an overview of the following sections of the manuscript. 

- The opening paragraph is followed by the section on “Child health and development as related to family psychosocial burden and parenting”. It is mostly identical to the section originally submitted, except for the paragraph which describes results of the KiD 0-3 2015 study, to which we added more detail (see Reviewer 1’s comment Nr 4).

The last section of the introduction now focuses on current topics, which can be assumed to have (recently had) an impact on families’ lives in Germany: the COVID-19 pandemic and the war in Ukraine. Since we aimed to move the focus from the experience of the pandemic to a better understanding on needs and resources of families in Germany now in more general terms, we deleted some details related to earlier research on associations between the pandemic and families’ psychosocial burden and child development. However, we did add more detail to the pandemic situation in Germany during the time of data collection (lines 203-210 in the Manuscript with Track Changes, lines 141-148 in the Manuscript); also see our reply to Reviewer 1’s comment Nr 6). We also added details on how the Ukraine war affects the German public in general and families with small children in particular (respectively lines 211-223 and 149-161 in the Manuscript with and without Track Changes), since we agree with the reviewers that this topic needs more background information, especially for readers outside of Germany (also see our reply to reviewer’s comment 14). Additionally, we made some corresponding changes throughout the manuscript: 

- We changed the word “determine” to “explore” in the sentence “Additionally, it will be important to determine explore whether the expansion of Germany’s Early Childhood Intervention program between 2015 and 2022 led to changes in how many and which families know about and make use of the associated support measures” in the discussion (line 466, Manuscript with Track Changes, line 388 in the Manuscript). 

- We added details to the angle from which we assess the families experience of the pandemic to the discussion (lines 483-491 in the Manuscript with Track Changes, lines 405-413 in the Manuscript), and why we consider this angle to be important. 

3. In the analysis section, please explain which analytic methods will be used to test the aims/make comparisons. Please also explain how missing data will be addressed for the different sets of analysis. I note that, if the study is reframed as in (1) above, descriptive analyses would be sufficient to address the aims.

 We added information on how missing data will be addressed in the analysis section, and on analytic methods that we plan to use in order to answer the different research questions. 

Other comments:

4. Line 104: I don’t understand this paragraph: different types of burden and stress are used multiple times and I can’t tell how they are different between the groups.

 We rewrote the paragraph and hope that this enhanced comprehensibility (lines 126-1381 in the Manuscript with Track Changes, lines 101-111 in the Manuscript). 

5. Line 123: “research has demonstrated changes” – this reads as though they are sustained changes, but this concept goes beyond the scope of this paper and the study cited was from the early months of the pandemic.

 Again, we would like to thank the reviewer for her very thorough reading! The sentence in question has been deleted in reorganizing and rewriting the introduction. 

6. Line 156: the phrasing “in the face of the COVID-19 pandemic” – it is not possible to judge whether this is current for German families or not, without more information in the Introduction. I have assumed like many countries, we’re in a ‘post-COVID-19’ world, in the sense that lockdowns have finished, people are vaccinated, and there’s a lot of COVID-19 circulating.

 The phrasing “in the face of the COVID-19 pandemic” was deleted as part of reframing the manuscript as suggested by this reviewer and outlined above. To enable readers of the manuscript to better understand the COVID-19 situation in Germany at the time of data collection, we added details in the introduction ((lines 203-210 in the Manuscript with Track Changes, lines 141-148 in the Manuscript)). We also added some sentences regarding the assessment of the COVID-19 pandemic to the discussion (lines 483-491 in the Manuscript with Track Changes, lines 405-413 in the Manuscript). 

7. Line 193: when is consent invited?

 Parental consent is invited after practice staff (and/or the pediatrician) informed the parents about the study and before the parents start to fill out the questionnaire. We added this information to the sentence in question (lines 267-269 in the Manuscript with Track Changes, lines 191-193 in the Manuscript).

8. Line 202: The 15% uptake through pediatricians is concerning, though understandable given the high research requirements in the face of overburdened health systems. I see that weighting will be used to approximate the German population; are there any risks to consider here around the representativeness of those attending pediatric offices and will this design need to consider those who don’t attend?

Indeed, a response rate of 15 percent is not particularly high. We based this calculation on the experience of the previous study, where a 15 % response rate was achieved. In-depth non-responder analyses of the 2015 data showed that a sufficient distribution was achieved with regard to the selection criteria. The majority of pediatricians who declined to participate gave shortness of time as the reason. In KiD 0-3 2022, we will also conduct in-depth analyses on the distribution of and, if necessary, include these in the weighting strategy. We added this information in the section “Sample size”. In addition, in the discussion (lines 448-451 in the Manuscript with Track Changes, lines 370-373 in the Manuscript)), we specify, that when we say “nearly all families in Germany make use of the well-child visits”, the number that has been reported in the literature referenced (Schmidtke et al 2018) is as high as 99 %. 

9. Line 199: Has recruitment already finished (November 2022)?

 Yes, recruitment finished in November, with the last families completing the online questionnaire in early December 2022. 

10. Table 1: why did the authors design measures for the pandemic experience instead of using questions from other well-known studies, like the UK Young Minds Matters or Australia’s National Child Health Poll or CRISIS questionnaire?

 We did our best to research existing measures for the pandemic experience, and could (with few exceptions for single items) not find measures, that we found entirely fitting for our study. Unfortunately, we must admit the we did not come across the measures suggested by this reviewer in our search. However, since we pretested the questionnaire with psychosocially burdened mothers, as described in the section on “Patient and public involvement”, we are confident, that the items we designed are relevant to and easily understandable by parents with young children in Germany. 

11. How feasible is the 25-minute questionnaire in clinic, plus the administrative parts, e.g. are clients waiting that long and if they are, will embedding this research increase these times (i.e. if the pediatrician must also enter data)?

Thank you for this question. Before the Covid-19 pandemic, waiting times at doctors' offices were quite a long time, so 25 minutes seemed reasonable. In the previous study in 2015, the similarly long questionnaire could also be completed well. In addition, we ensured in the study design that the processes were as clear as possible through training and an appealing design of the study documents. Furthermore, it is possible to interrupt the questionnaire and finish it later - e.g. at home. The pretest also showed that parents were able to complete the questionnaire within 25 minutes.

How the pediatricians and the practice team organized the process in detail in their practices to keep the effort as low as possible was discussed during the training sessions. The practices were as flexible as necessary to optimize the processes according to their routines.

12. How much would a pediatrician be paid for a standard consult, and how does the 60 euro compare? This also helps understand feasibility.

 Depending on the state in which the practice is located, pediatricians are paid between 50-60 € for a standard well-child screening. The amount of the incentive has been decided upon in consultation with the Professional Association of Pediatricians (Berufsverband der Kinder- und Jugendärzte, BVKJ). We agree with the reviewer, that this information helps readers understand the feasibility if the study and added information (lines 372-373 in the Manuscript with Track Changes, lines 296-297 in the Manuscript). 

13. Line 376: first data – please add ‘In Germany’.

 We added “in Germany” to the sentence so it now reads: “Also, to our knowledge, our study will be the first to provide data on the wellbeing of young children and their families in Germany since the COVID-19 pandemic in a nationally representative sample” (lines 473-474 in the Manuscript with Track Changes, lines 395-396 in the Manuscript)

14. Line 377: The aspects on the experience of the Ukraine war need more justification if you want to include them as it was a surprise to see them here.

 We fully agree that the aspects on the experience of the Ukraine war must have come as a surprise and apologize for the oversight on our part. We added information to the introduction (respectively lines 211-223 and 149-161 in the Manuscript with and without Track Changes). 

15. The PLOS ONE data sharing for Protocols asks: “For protocols without pilot or preliminary data, authors are strongly encouraged to state how they plan to share research data from their study when it is completed or published.” Can you please add something to this?

 As stated in response to the editor, we would like to include the data availability statement as follows: “The data in this study is collected from patients and their physicians. The patients provide informed consent to release their medical information for the purpose of our analysis. However, to protect the privacy and confidentiality of the patients, the data is not publicly available. The authors confirm that all data necessary to support the findings of this study are available upon request to the corresponding author. Requests for access to the data should be made to the corresponding author and will be subject to ethical review and approval.”

7. PLOS authors have the option to publish the peer review history of their article (what does this mean?). If published, this will include your full peer review and any attached files.

Do you want your identity to be public for this peer review? For information about this choice, including consent withdrawal, please see our Privacy Policy.

Reviewer #1: Yes: Anna Price

---

## [Decision Letter · Decision Letter 1]

2 May 2023

How are families in Germany doing in 2022? Study Protocol of a nationally representative, cross-sectional study of parents with children aged 0-3 years

PONE-D-22-25670R1

Dear Dr. Neumann,

We’re pleased to inform you that your manuscript has been judged scientifically suitable for publication and will be formally accepted for publication once it meets all outstanding technical requirements.

Kind regards,

Humayun Kabir, MSc in Epidemiology

Academic Editor

PLOS ONE

Additional Editor Comments (optional):

Reviewers' comments:

Reviewer's Responses to Questions

**Comments to the Author**

1. Does the manuscript provide a valid rationale for the proposed study, with clearly identified and justified research questions?

Reviewer #1: Yes

2. Is the protocol technically sound and planned in a manner that will lead to a meaningful outcome and allow testing the stated hypotheses?

Reviewer #1: Yes

3. Is the methodology feasible and described in sufficient detail to allow the work to be replicable?

Reviewer #1: Yes

4. Have the authors described where all data underlying the findings will be made available when the study is complete?

Reviewer #1: Yes

5. Is the manuscript presented in an intelligible fashion and written in standard English?

Reviewer #1: Yes

6. Review Comments to the Author

You may also provide optional suggestions and comments to authors that they might find helpful in planning their study.

Reviewer #1: Thank you for the opportunity to re-review this manuscript, which is much improved. I recognize the substantial and careful changes you have made, and appreciate your considered and enthusiastic responses to my original suggestions. I wish you all the best for your important work.

7. PLOS authors have the option to publish the peer review history of their article (what does this mean?). If published, this will include your full peer review and any attached files.

Reviewer #1: **Yes: **Anna Price

<quillbot-extension-portal></quillbot-extension-portal><quillbot-extension-portal></quillbot-extension-portal>

---

## [Editor Report · Acceptance letter]

5 May 2023

PONE-D-22-25670R1 

How are families in Germany doing in 2022? Study Protocol of a nationally representative, cross-sectional study of parents with children aged 0-3 years 

Dear Dr. Neumann:

I'm pleased to inform you that your manuscript has been deemed suitable for publication in PLOS ONE. Congratulations! Your manuscript is now with our production department. 

Kind regards, 

on behalf of

Dr. Humayun Kabir 

Academic Editor

PLOS ONE